# Structural Smoothing Low-Rank Matrix Restoration Based on Sparse Coding and Dual-Weighted Model

**DOI:** 10.3390/e24070946

**Published:** 2022-07-07

**Authors:** Jiawei Wu, Hengyou Wang

**Affiliations:** School of Science, Beijing University of Civil Engineering and Architecture, Beijing 100044, China; 2107010420009@stu.bucea.edu.cn

**Keywords:** group sparse coding, low-rank regularized group sparse coding, TV norm, dual-weighted, image restoration

## Abstract

Group sparse coding (GSC) uses the non-local similarity of images as constraints, which can fully exploit the structure and group sparse features of images. However, it only imposes the sparsity on the group coefficients, which limits the effectiveness of reconstructing real images. Low-rank regularized group sparse coding (LR-GSC) reduces this gap by imposing low-rankness on the group sparse coefficients. However, due to the use of non-local similarity, the edges and details of the images are over-smoothed, resulting in the blocking artifact of the images. In this paper, we propose a low-rank matrix restoration model based on sparse coding and dual weighting. In addition, total variation (TV) regularization is integrated into the proposed model to maintain local structure smoothness and edge features. Finally, to solve the problem of the proposed optimization, an optimization method is developed based on the alternating direction method. Extensive experimental results show that the proposed SDWLR-GSC algorithm outperforms state-of-the-art algorithms for image restoration when the images have large and sparse noise, such as salt and pepper noise.

## 1. Introduction

Sparse representation theory [1,2,3] has always been a very interesting research field due to its good performance in the fields of computer vision and image processing. It can be well used to represent or extract the main features of images. Among existing works, the image denoising methods based on patch sparse coding [4] divide the image into each patch equally, and use the patch’s structure to encode the image into a combination of dictionary and sparse coefficient. By constraining the $L_{0}$ norm of coding coefficients, the real coding can be approximately estimated on the noisy image, so as to remove the noise.

On this topic, the dictionary is very important for the performance of these methods and should be learned firstly. Thus, the popular technologies such as PCA [5], K-SVD [6], and S-KSVD [7] are used to train dictionaries, to achieve higher expressive ability. However, they are all large-scale and highly non-convex problems, which often have high computational complexity. On the other hand, patches are units of sparse representation. Each patch is usually considered independently in dictionary learning and sparse coding, which leads to its focus on the local structure of the image and essentially ignores the relationship between similar patches—that is, the non-local self similarity (NSS) of the image [8,9,10,11].

For these non-local image denoising approaches, they provide a new research theory and direction for image denoising. Group sparse coding (GSC) [12], which uses groups instead of a single patch as the basic unit of sparse coding, combines the advantages of local sparsity and NSS of images, and shows great potential in various image processing tasks [13,14]. Similar to the sparse representation based on patches, each image patch group can also be accurately fitted by the sparse linear combination of dictionary atoms. In order to make full use of the similarity between groups, Zha et al. proposed low-rank regulated group sparse coding (LR GSC) [15], which imposed a low-rank constraint on the sparse coefficients of each group. It is the first time that the low-rank property of group sparsity in GSC has been used to ensure that the dictionary domain coefficients are not only sparse but also low-rank. Although the above methods have achieved good results, they are mainly aimed at the removal of Gaussian white noise, and cannot be used to remove the outlier noise effectively. In addition, when the density of noise is large, the acquisition of similar patches will be affected since each image patch also contains these noisy pixels. Figure 1 shows the effect of the LR-GSC model on image restoration from Gaussian noise and outlier noise, respectively. It can be seen that the LR-GSC model is more suitable for removing Gaussian noise than outlier noise.

Because of the two-dimensional structure of a matrix, it can effectively protect the original details and structure information of the images. It is also more robust to data with outliers—for example, salt and pepper noise. Wright et al. [16] proposed the problem of low-rank restoration, which decomposes the original data as the sum of the low-rank matrix and sparse noise matrix. Candes et al. [17] accurately removed large low-rank matrices containing noise samples through kernel norm minimization. Using this theory to denoise the image, the original image structure with low-rank characteristics is separated from the observed noisy image, which has good performance on the removal of impulse and outlier noise.

When the rank or sparsity exceeds the threshold limit, the convex approximation model will not be able to accurately estimate the low-rank solution and sparse solution. For this reason, Candes et al. [18] proposed a weighted $L_{1}$ norm method to allocate smaller weights to larger matrix elements, restrain the overshrinkage of the $L_{1}$ norm, and improve the accuracy of the sparse solution. Gu et al. [19] proposed the weighted nuclear norm minimization (WNNM) model and the WNNM-based RPCA model (WNNM-RPCA) [20]. The model uses the weighted nuclear norm to relax the rank function. Considering the importance of different rank components, they assign weights according to the size of singular values to control the penalization of different rank components, so as to retain more important rank components and improve the accuracy of low-rank solutions. Peng et al. [21] proposed a double-weighted model, which simultaneously weighted the sparse and low-rank terms in the RPCA model, and was combined with the reweighted method to allocate weights in the iterative algorithm. The model can improve the accuracy of the sparse solution and low-rank solution at the same time. Although the low-rank theory performs well in removing impulse or outlier noise, it is mainly based on local similarity—that is, only the overall low-rank structure is considered.

Therefore, in order to overcome the above shortcomings, we propose a smoothing dual-weighted low-rank group sparse coding (SDWLR-GSC) algorithm to remove impulse or outlier noise. The main contributions are as follows:

(1) In addition to using the non-local self-similarity prior to the image, as with LR-GSC, to maintain the low rank of similar patch coefficients when constructing similar patches, we also impose a low-rank constraint on the reconstructed similar patches as a whole. We not only consider the low-rank property between similar patches, but also consider local structural smoothness. Based on this, we combine the model with the TV norm to further strengthen the local structural smoothness of the matrix. Figure 2 shows an example of image recovery from dense noise based on using and not using the TV norm.

(2) Because the low rank of the coefficients of similar patches and the global low rank of the reconstruction matrix of similar patches are both considered in the objective function, the optimization problem becomes a challenging non-convex optimization problem. In order to solve this problem effectively, we develop a new numerical solution based on the inexact augmented Lagrange multiplier (IALM) and non-uniform singular value threshold (NSVT). The experimental results show that the proposed method can improve the quality of the restored image significantly.

The remainder of this paper is organized as follows. Section 2 reviews the related work on low-rank regularized group sparse coding (LR-GSC). Section 3 presents our proposed SDWLR-GSC model, and Section 4 shows the solution of our proposed method. The experimental results are presented in Section 5. Finally, Section 6 summarizes this paper.

## 2. Related Work

In this section, we will review the work related to our proposed method.

### 2.1. Low-Rank Regularized Group Sparse Coding

In GSC [13,14], the method is mainly divided into two steps: grouping similar patches, and then sparsely representing the grouped patches, which can be formulated as the following problem,
(1)$${\hat{A}}_{i}=arg\underset{A_{i}}{min}(\frac{1}{2}∥{X_{i}-D_{i}A_{i}∥}_{F}^{2}+\lambda ∥A_{i}{∥}_{1})\;\forall i$$
where $X_{i}\in R^{b\times m}$ is a group composed of patches with similar structure, and $D_{i}$ is the dictionary learned from each group of $X_{i}$. ${∥\cdot ∥}_{F}$ and ${∥\cdot ∥}_{1}$ denote the Frobenius norm and $L_{1}$ norm, respectively. Thus, the optimal ${\hat{A}}_{i}$ is the sparse codes of $X_{i}$ with λ≥0.

Zha et al. [15] proposed LR-GSC, which combines the low-rank characteristics of group coefficients in GSC, to further utilize the similarity between groups; the dictionary domain coefficients of each group are constrained as not only sparse but also low-rank. Taking the sparsity penalty and low-rank penalty as dual regularizers, the following problem is obtained: (2)$$\begin{array}{cc} \{{\hat{A}}_{i},{\hat{B}}_{i}\}= & arg{min}_{A_{i},B_{i}}\frac{1}{2}∥{X_{i}-D_{i}A_{i}∥}_{F}^{2}+\lambda {∥A_{i}∥}_{1}\\ \; & \;\;+\frac{1}{\eta }∥{A_{i}-B_{i}∥}_{F}^{2}+\tau ∥{B_{i}∥}_{*}\;\forall i, \end{array}$$
where ${∥\cdot ∥}_{1}$ is applied for the sparsity penalty, and the nuclear norm ${∥\cdot ∥}_{*}$ is applied for the low-rank penalty. τ is a non-negative constant, and η is a balancing factor in making (Equation 2) more feasible. A low-rank approximation $B_{i}$ is jointly estimated for each group sparse matrix $A_{i}$. Similar to GSC, the optimal sparse codes ${\left\{{\hat{A}}_{i}\right\}}_{i=1}^{n}$ are used to restore the latent image.

### 2.2. Dual-Weighted Low-Rank Matrix Recovery

The authors in [22] proved that the RPCA model can be transformed into solving the following convex optimization problem by minimizing the combined decomposition matrix of the $L_{1}$ norm and nuclear norm, i.e.,
(3)$$\begin{array}{cc} \; & {min}_{X,S}{∥X∥}_{*}+\lambda {∥S∥}_{1},\\ \; & s.t.\;X+S=Y, \end{array}$$
where $Y\in M^{m\times n}$ is the observed matrix. ${∥X∥}_{*}$ is the nuclear norm of matrix *X*, which is defined as the sum of matrix singular values, i.e., ${∥X∥}_{*}=\sum_{i=1}^{r}\sigma_{i}\left(X\right)$, r=min(m,n), $\sigma_{i}\left(X\right)$ is the *i*-th singular value of matrix *X*. ${∥S∥}_{1}$ is the $L_{1}$ norm of matrix *S*, which is defined as the sum of matrix absolute elements—that is, ${∥S∥}_{1}=\sum_{i=1}^{m}\sum_{j=1}^{n}\left|s_{i,j}\right|$, $s_{i,j}$ represents the elements in matrix *S*. λ>0 is the regularization parameter.

In the process of solving, the model can easily lead to overshrinkage, which affects the accuracy of the solution. In [21], the sparse term and low-rank term in the RPCA model are weighted at the same time, and a double-weighted model is proposed, i.e.,
(4)$$\begin{array}{cc} \; & {min}_{X,S}∥{X∥}_{\Omega ,*}+\lambda {∥W⊙S∥}_{1},\\ \; & s.t.\;X+S=Y, \end{array}$$
where $∥{X∥}_{\Omega ,*}=\sum_{i=1}^{r}w_{i}\sigma_{i}\left(X\right)$, ${∥W⊙S∥}_{1}=\sum_{i=1}^{m}\sum_{j=1}^{n}w_{i,j}\left|s_{i,j}\right|$, $s_{i,j}$ represents the elements in matrix *S*, $W\in R^{m\times n}$ is the weight matrix. The weight of $s_{i,j}$ is defined as $w_{i,j}=1/\left(|s_{i,j}|+\varepsilon \right)$.

## 3. Formulation of the Proposed Method

Low-rank regularized group sparse coding (LR-GSC) [15] ensures that the dictionary domain coefficients of each group are not only sparse but also low-rank. However, when the density of outlier noise becomes higher, the method cannot recover the image well. In this section, inspired by the low-rank matrix restoration algorithm in [21,23], we propose a new group sparse coding model, called SDWLR-GSC, which introduces total variation (TV) regularization into low-rank group sparse coding to realize image structure smoothing. At the same time, the dual-weighted model is used to recover the clean image better from the large and sparse noise. Our new method can be formulated as follows: (5)$$\begin{array}{cc} \; & {min}_{X_{i},S_{i},A_{i}}\sum_{j=1}^{q}w_{X_{i},j}\cdot {\tilde{\sigma }}_{j}+\theta ∥W_{S_{i}}⊙S_{i}{∥}_{1}+\beta {∥X_{i}∥}_{TV}\\ \; & \;\;+\frac{1}{2\rho }\sum_{i=1}^{n}∥Y_{i}-D_{i}A_{i}{∥}_{F}^{2}+\lambda \sum_{i=1}^{n}∥A_{i}{∥}_{1}+\tau \sum_{i=1}^{n}{∥A_{i}∥}_{*}\\ \; & s.t.\;X_{i}\in B_{l,u}\equiv \left\{x_{i,j},\;l\leq x_{i,j}\leq u\right\},\\ \; & \;\;{\hat{Y}}_{i}=X_{i}+S_{i}. \end{array}$$
where $Y_{i}\in R^{b\times m}$ represents the matrix constructed by the batches of similar patches for each basic patch $y_{i}$. ${\hat{Y}}_{i}$ is calculated by multiplying ${\hat{D}}_{i}$ and ${\hat{A}}_{i}$. ${\hat{D}}_{i}$ and ${\hat{A}}_{i}$ are the dictionary learned from each group $Y_{i}$ and the sparse representation of each group $X_{i}$ under a given dictionary, respectively. $W_{X_{i}}=\left\{w_{X_{i},j}\right\}$ and $W_{S_{i}}\in R^{m\times n}$ are weights of $\left\{{\tilde{\sigma }}_{j}\right\}$ and $S_{i}$. $\left\{{\tilde{\sigma }}_{j}\right\}$ are singular values of matrix $X_{i}$. The constraint $B_{l,u}$ states that pixel values are bounded—for example, [0, 255]. ${∥\cdot ∥}_{TV}$ denotes the TV norm. In [24], total variation is proposed as a penalty term to deal with image restoration. The advantage of TV regularization is that it can well restore the edge and eliminate the noise.

Apparently, the proposed SDWLR-GSC, based on LR-GSC, exploits the sparsity and low-rankness of the dictionary domain coefficients of each group at the same time. For the image patches reconstructed by the given dictionary and the sparse coding, we further introduce the dual-weighted model to weight low-rank terms and sparse terms, and we add the TV norm to maintain the smoothing structure of images. Therefore, the proposed model can better separate the clean image from the sparse noise and achieve better reconstruction results. Figure 3 shows the flowchart of the proposed SDWLR-GSC by taking the simple image denoising as an example.

## 4. The Proposed Solution

In this section, to solve the proposed SDWLR-GSC problem, we will present our solution method based on the alternating direction method of multipliers, which separates the multi-variable optimal problem into several single-variable problems, so that these variables can be undated one by one in each iteration.

First, to address the problem more easily, we introduce an auxiliary variable as follows: (6)$$\begin{array}{cc} \; & {min}_{H_{i},X_{i},S_{i},A_{i},B_{i}}\sum_{j=1}^{q}w_{H_{i},j}\cdot \sigma_{j}+\theta ∥W_{S_{i}}⊙S_{i}{∥}_{1}+\beta {∥X_{i}∥}_{TV}\\ \; & \;\;+\frac{1}{2\rho }\sum_{i=1}^{n}∥Y_{i}-D_{i}A_{i}{∥}_{F}^{2}+\lambda \sum_{i=1}^{n}∥A_{i}{∥}_{1}+\tau \sum_{i=1}^{n}{∥B_{i}∥}_{*}\\ \; & s.t.\;X_{i}\in B_{l,u}\equiv \left\{x_{i,j},l\leq x_{i,j}\leq u\right\},\\ \; & \;\;{\hat{Y}}_{i}=X_{i}+S_{i},H_{i}=X_{i},B_{i}=A_{i}. \end{array}$$
where $W_{H_{i}}=\left\{w_{H_{i},j}\right\}$ are weights for $\left\{{\tilde{\sigma }}_{j}\right\}$, $\left\{{\tilde{\sigma }}_{j}\right\}$ are singular values of matrix $H_{i}$, and $w_{H_{i},j}=w_{X_{i},j}$, j=1,…,n.

In this way, the augmented Lagrange function of (Equation 7) can be obtained as follows: (7)$$\begin{array}{cc} \; & f(H_{i},X_{i},S_{i},A_{i},B_{i},Y_{1},Y_{2})\\ \; & \;=\sum_{j=1}^{n}w_{H_{i},j}\cdot \sigma_{j}+\theta ∥W_{S_{i}}⊙S_{i}{∥}_{1}+\beta {∥X_{i}∥}_{TV}\\ \; & \;\;+\frac{1}{2\rho }\sum_{i=1}^{m}∥Y_{i}-D_{i}A_{i}{∥}_{F}^{2}+\lambda \sum_{i=1}^{n}{∥A_{i}∥}_{1}\\ \; & \;\;+\frac{1}{2\eta }\sum_{i=1}^{m}∥A_{i}-B_{i}{∥}_{F}^{2}+\tau \sum_{i=1}^{n}{∥B_{i}∥}_{*}\\ \; & \;\;+\langle Y_{1},Y_{i}-H_{i}-S_{i}\rangle +\langle Y_{2},X_{i}-H_{i}\rangle \\ \; & \;\;+\frac{\mu }{2}\left({∥Y_{i}-H_{i}-S_{i}∥}_{F}^{2}+{∥X_{i}-H_{i}∥}_{F}^{2}\right)\\ \; & s.t.\;X_{i}\in B_{l,u}\equiv \left\{x_{i,j},l\leq x_{i,j}\leq u\right\}. \end{array}$$

Then, we use the iterative alternating direction method, which optimizes one variable and fixes the remaining optimization variables in an iterative way. In this way, the original complex multi-variable optimization problem can be simplified to a single-variable optimization problem, and its solution can be obtained analytically. In our problem, there are seven sub-problems, i.e., $H_{i}$, $X_{i}$, $S_{i}$, $A_{i}$, $B_{i}$, $Y_{1}$, $Y_{2}$ sub-problems. Next, we will give a detailed implementation for each of them.

### 4.1. $H_{i}$ Sub-Problem

If we fix the variables $\left(X_{i},S_{i},A_{i},B_{i},Y_{1},Y_{2}\right)$, the variable $H_{i}$ can be solved by minimizing $f(H_{i},X_{i},S_{i},A_{i},B_{i},Y_{1},Y_{2})$. Specifically,
(8)$$\begin{array}{cc} \; & arg{min}_{H_{i}}f\left(H_{i},X_{i},S_{i},A_{i},B_{i},Y_{1},Y_{2}\right)\\ \; & \;\;=arg{min}_{H_{i}}\sum_{j=1}^{n}w_{H_{i},j}\cdot \sigma_{j}\\ \; & \;\;+\langle Y_{1},Y_{i}-H_{i}-S_{i}\rangle +\langle Y_{2},X_{i}-H_{i}\rangle \\ \; & \;\;+\frac{\mu }{2}\left({∥Y_{i}-H_{i}-S_{i}∥}_{F}^{2}+{∥X_{i}-H_{i}∥}_{F}^{2}\right)\\ \; & \;\;=arg{min}_{H_{i}}\sum_{j=1}^{n}w_{H_{i},j}\cdot \sigma_{j}\\ \; & \;\;+\mu ∥H_{i}-\frac{1}{2}\left(Y_{i}+X_{i}-S_{i}+Y_{1}/\mu +Y_{2}/\mu \right){∥}_{F}^{2}\\ \; & \;\;=arg{min}_{H_{i}}\sum_{j=1}^{n}w_{H_{i},j}\cdot \sigma_{j}+\mu {∥H_{i}-L∥}_{F}^{2}, \end{array}$$
where $L=\frac{1}{2}\left(Y_{i}+X_{i}-S_{i}+Y_{1}/\mu +Y_{2}/\mu \right)$. This minimization problem (Equation 8) can be solved by non-uniform singular value thresholding (NSVT) [23] as follows: (9)$$H=D_{{\left(2\mu \right)}^{-1}W_{H}}\left(L\right).$$
where $D_{\epsilon }\left(Q\right)=US_{\epsilon }\left(\Sigma \right)V^{T}$, $U\Sigma V^{T}$ is the singular value decomposition of matrix *Q*. $S_{\epsilon }\left(\cdot \right)$ denotes non-uniform soft thresholding operator [25], and its (i,j)-th element is $max(|q_{ij}\left|-\epsilon ,0\right)sgn\left(q_{ij}\right)$, where the parameter ϵ>0.

### 4.2. $X_{i}$ Sub-Problem

If the variables $\left(H_{i},S_{i},A_{i},B_{i},Y_{1},Y_{2}\right)$ are fixed, we can optimize $X_{i}$ by minimizing $f(H_{i},X_{i},S_{i},A_{i},B_{i},Y_{1},Y_{2})$. Specifically,
(10)$$\begin{array}{cc} \; & arg{min}_{X_{i}}f\left(H_{i},X_{i},S_{i},A_{i},B_{i},Y_{1},Y_{2}\right)\\ \; & \;\;=arg{min}_{X_{i}}\beta ∥X_{i}{∥}_{TV}+\langle Y_{2},X_{i}-H_{i}\rangle +\frac{\mu }{2}{∥X_{i}-H_{i}∥}_{F}^{2}\\ \; & \;\;=arg{min}_{X_{i}}\beta ∥X_{i}{∥}_{TV}+\frac{\mu }{2}{∥X_{i}-\left(H_{i}-Y_{2}/\mu \right)∥}_{F}^{2}\\ \; & \;\;=arg{min}_{X_{i}}\beta ∥X_{i}{∥}_{TV}+\frac{\mu }{2}{∥X_{i}-R∥}_{F}^{2}, \end{array}$$
where $R=(H_{i}-Y_{2}/\mu )$, $X_{i}\in B_{l,u}\equiv \left\{x_{i,j},l\leq x_{i,j}\leq u\right\}$. Similar to the constrained convex problem represented by the image denoising problem using the TV norm in [23], if μ and β are given, γ can be obtained by β/μ. Then, the solution of problem (Equation 10) is given by: (11)$$X_{i}=P_{B_{l,u}}\left(R-\gamma L(p,q)\right).$$
where (p,q), *L*, *P* are the matrix pairs, linear operator, and orthogonal projection operator, respectively. In addition, the bounds of the constraint are set to be [l,u] = [0, 255].

### 4.3. $S_{i}$ Sub-Problem

Similar to other variables, if we fix the variables $\left(H_{i},X_{i},A_{i},B_{i},Y_{1},Y_{2}\right)$, the variable $S_{i}$ can be updated by the optimal solution of minimizing $f(H_{i},X_{i},S_{i},A_{i},B_{i},Y_{1},Y_{2})$.
(12)$$\begin{array}{cc} \; & arg{min}_{S_{i}}f\left(H_{i},X_{i},S_{i},A_{i},B_{i},Y_{1},Y_{2}\right)\\ \; & \;\;=arg{min}_{S_{i}}\theta {∥W_{S_{i}}⊙S_{i}∥}_{1}+\langle Y_{1},Y_{i}-H_{i}-S_{i}\rangle \\ \; & \;\;+\frac{\mu }{2}{∥Y_{i}-H_{i}-S_{i}∥}_{F}^{2}\\ \; & \;\;=arg{min}_{S_{i}}\theta {∥W_{S_{i}}⊙S_{i}∥}_{1}\\ \; & \;\;+\frac{\mu }{2}{∥S_{i}-\left(Y_{i}-H_{i}+Y_{2}/\mu \right)∥}_{F}^{2}, \end{array}$$

This minimization problem (Equation 12) can be solved by non-uniform soft thresholding (NST) [25] as follows: (13)$$S=S_{\theta \mu^{-1}W_{S}}\left(Y-H+Y_{1}/\mu \right).$$

### 4.4. $A_{i}$ Sub-Problem

If we fix the variables $\left(H_{i},S_{i},X_{i},B_{i},Y_{1},Y_{2}\right)$, the $A_{i}$ sub-problem can be reduced to the following optimal problem.
(14)$$\begin{array}{cc} \; & arg{min}_{A_{i}}f\left(H_{i},X_{i},S_{i},A_{i},B_{i},Y_{1},Y_{2}\right)\\ \; & \;=\frac{1}{2\rho }\sum_{i=1}^{n}{∥Y_{i}-D_{i}A_{i}∥}_{F}^{2}\\ \; & \;\;+\frac{1}{2\eta }\sum_{i=1}^{n}∥A_{i}-B_{i}{∥}_{F}^{2}+\lambda \sum_{i=1}^{n}{∥A_{i}∥}_{1} \end{array}$$

We use the principal component analysis (PCA) of each group to learn the grouping sub-dictionary $D_{i}$ [26]. Since the learned PCA dictionary is singular, Equation (Equation 14) is equivalent to the following problem.
(15)$$\begin{array}{cc} \; & arg{min}_{A_{i}}f\left(H_{i},X_{i},S_{i},A_{i},B_{i},Y_{1},Y_{2}\right)\\ \; & \;=arg{min}_{A_{i}}\frac{1}{2\rho }\sum_{i=1}^{n}{∥G_{i}-A_{i}∥}_{F}^{2}\\ \; & \;\;+\frac{1}{2\eta }\sum_{i=1}^{n}∥A_{i}-B_{i}{∥}_{F}^{2}+\lambda \sum_{i=1}^{n}{∥A_{i}∥}_{1}\\ \; & \;=arg{min}_{A_{i}}\frac{1}{2}\sum_{i=1}^{n}∥P_{i}-A_{i}{∥}_{F}^{2}+v\sum_{i=1}^{n}{∥A_{i}∥}_{1}, \end{array}$$
where $Y_{i}≜D_{i}G_{i}$, $P_{i}≜\frac{\eta G_{i}+\rho B_{i}}{\eta +\rho }$ and v=ρηλ. Then, this minimization problem (Equation 15) can be solved by non-uniform soft thresholding (NST) [25] as follows: (16)$$S=S_{v}\left(P_{i}\right)\;\forall i.$$

### 4.5. $B_{i}$ Sub-Problem

The same as the above methods, if we want to find the updating formula of $B_{i}$, the variables $\left(H_{i},S_{i},X_{i},A_{i},Y_{1},Y_{2}\right)$ should also be fixed firstly, and then the solution of the variable $B_{i}$ can be obtained by minimizing $f(H_{i},X_{i},S_{i},A_{i},B_{i},Y_{1},Y_{2})$. Specifically,
(17)$$\begin{array}{cc} \; & arg{min}_{B_{i}}f\left(H_{i},X_{i},S_{i},A_{i},B_{i},Y_{1},Y_{2}\right)\\ \; & \;=arg{min}_{B_{i}}\frac{1}{2\eta }∥A_{i}-B_{i}{∥}_{F}^{2}+\tau {∥B_{i}∥}_{*} \end{array}$$

This minimization problem (Equation 17) can be solved by non-uniform singular value thresholding (NSVT) [23] as follows: (18)$$B_{i}=D_{\eta \tau }\left(A_{i}\right).$$

### 4.6. $Y_{1}$ and $Y_{2}$ Sub-Problems

Finally, $Y_{1}$ and $Y_{2}$ are Lagrange multiplier matrices of the original optimization problem. They should be updated after other variables. If $Y_{1}$ is unknown and other variables are fixed, $Y_{1}$ can be updated as follows: (19)$$Y_{1}=Y_{1}+\mu \left(Y-H-S\right).$$

If $Y_{2}$ is unknown and other variables are fixed, $Y_{2}$ can be updated as follows: (20)$$Y_{2}=Y_{2}+\mu \left(X-H\right).$$

In Algorithm 1, we summarize the complete algorithm of SDWLR-GSC for image restoration.
**Algorithm 1** Smoothed and Dual-Weighted Low-Rank Group Sparse Coding (SDWLR-GSC).**Require:** The degraded image $y\in R^{m\times n}$ (assuming m≥n);**Output:** The final restored image ${\hat{X}}^{\left(K\right)}$;_1:_  **Initialization:** ${\hat{x}}^{\left(0\right)}=y$, $A_{i}^{\left(0\right)}=0$, $B_{i}^{\left(0\right)}=0$;_2:_  Set parameters *b*, *m*, ρ, η, λ, τ and Max-Iter;_3:_  **for** k=1 **to** Max-Iter **do**_4:_    Iterative regularization $y^{\left(k\right)}={\hat{x}}^{\left(k-1\right)}+\delta \left(y-{\hat{y}}^{\left(k\right)}\right)$_5:_    Divide $y^{\left(k\right)}$ into a set of overlapping patches with size b×b_6:_    **for** each patch $y_{i}$ in $y^{\left(k\right)}$ **do**_7:_       Find non-local similar patches to form a group $Y_{i}$_8:_       Construct dictionary $D_{i}$ by $Y_{i}$ using PCA_9:_       Update $A_{i}$ by computing $A_{i}=D_{i}^{T}Y_{i}$_10:_     Perform $\left[U_{i},\Delta_{i},V_{i}\right]=SVD\left(A_{i}\right)$_11:_     Estimate ${\hat{B}}_{i}$ by computing Equation (Equation 18)._12:_     Estimate ${\hat{A}}_{i}$ by computing Equation (Equation 16)._13:_     Update ${\hat{Y}}_{i}$ by computing ${\hat{Y}}_{i}=D_{i}A_{i}$_14:_     Use **Algorithm 2** to ${\hat{Y}}_{i}$ to estimate $X_{i}$_15:_   **end for**_16:_   Aggregate $X_{i}$ to form the clean image ${\hat{X}}^{\left(k\right)}$_17:_ **end for**

**Algorithm 2** Smoothed and Dual-Weighted Model for Image Denoising.
**Require:** non-local similar patched group $Y_{i}\in R^{p\times q}$ (assuming p≥q);
**Output:** solutions $X_{i}^{\left(k\right)}=X_{i}^{\left(t+1\right)}$;
_1:_   **Initialization:** $W_{X_{i}}^{\left(0\right)}=1\in R^{q}$, $W_{S_{i}}^{\left(0\right)}=1\cdot 1^{T}\in R^{p\times q}$, $\left(X_{i}^{\left(0\right)},S_{i}^{\left(0\right)}\right)\in R^{p\times q}$, $H^{\left(0\right)}\in R^{p\times q}$, $Y_{1}^{\left(0\right)}\in R^{p\times q}$, $Y_{2}^{\left(0\right)}\in R^{p\times q}$;
_2:_   Set parameters $\mu_{0}>0$, $\xi =10^{-7}$, k=0, λ, δ and inneriter;
_3:_   **while**$∥Y_{i}-X_{i}-S_{i}{∥}_{F}/{∥Y_{i}∥}_{F}>\xi$ and t<inneriter **do**
_4:_       Let $L^{\left(t+1\right)}=Y_{i}+X_{i}^{\left(t\right)}-S_{i}^{\left(t\right)}+Y_{1}^{\left(t\right)}/\mu^{\left(t\right)}$, $\rho =\eta /\mu^{\left(t\right)}$, $H_{i}^{\left(t+1\right)}=D_{{\mu^{\left(t\right)}}^{-1}W_{X_{i}}}\left(L^{\left(t+1\right)}\right)$
_5:_       Let $R^{\left(t+1\right)}=H^{\left(t+1\right)}-Y_{2}^{\left(t\right)}$ using the **FGP Algorithm [23]** to compute $X_{i}^{\left(t+1\right)}=$
   $P_{B_{l,u}}\left(R^{\left(t+1\right)}-\rho L\left(p,q\right)\right)$
_6:_       $S^{\left(t+1\right)}=S_{\lambda {\mu^{\left(t\right)}}^{-1}W_{S_{i}}}(Y_{i}-H^{\left(t+1\right)}+Y_{1}^{\left(t\right)}/\mu^{\left(t\right)})$
_7:_       $Y_{1}^{\left(t+1\right)}=Y_{1}^{\left(t\right)}+\mu^{\left(t\right)}\left(Y_{i}-H^{\left(t+1\right)}-S_{i}^{\left(t+1\right)}\right)$
_8:_       $Y_{2}^{\left(t+1\right)}=Y_{2}^{\left(t\right)}\left(X_{i}^{\left(t+1\right)}-H^{\left(t+1\right)}\right)$
_9:_       $\mu^{\left(t+1\right)}=\delta \mu^{\left(t\right)},t\to t+1$
_10:_      Update weights ** [27]**: The weights for each i=1,⋯,p and j=1,⋯,q are updated
            by
                                          $w_{X_{i},j}^{\left(k+1\right)}=\frac{1}{\sigma_{j}^{\left(k\right)}+\epsilon_{X_{i}}}$, $w_{S_{i},ij}^{\left(k+1\right)}=\frac{1}{|{S_{i}}_{ij}^{\left(k\right)}|+\epsilon_{S_{i}}}$.
             where $\epsilon_{X_{i}}$ and $\epsilon_{S_{i}}$ are predetermined positive constants, and the singular value
             matrix
                                       $\Sigma^{\left(k\right)}=diag\left(\left[\sigma_{1}^{\left(k\right)},\cdots ,\sigma_{n}^{\left(k\right)}\right]\right)\in R^{n\times n}$
             with $\left[U^{\left(k\right)},\Sigma^{\left(k\right)},V^{\left(k\right)}\right]=svd\left({X_{i}}^{\left(k\right)}\right)$
_11:_ **end while**


## 5. Results

To verify the performance of the SDWLR-GSC algorithm in image restoration, we carried out a large number of experiments and compared its performance with the most novel methods based on the low-rank matrix. We also selected 16 classical images (Lena, Barbara, Couple, House, Monarch are from the Set12 dataset. Frame, Road, Bridge, Elaine, Pentagon, Lin are from Google Images. Flower is from the Kodak24 dataset. Monkey, Tank are from the USC-SIPI image dataset http://sipi.usc.edu/database/ (accessed on 28 March 2022)) with size 512×512 as the test dataset. The images used in all experiments are shown in Figure 4. This experiment mainly aimed at outlier noise with high density. Outlier noise includes two types: salt and pepper noise and random-valued sparse additive noise. Thus, we introduced two classes of numerical experiments. First, the test image was only destroyed by different levels of large and sparse additive noise. Second, the test image was only destroyed by different levels of salt and pepper noise. The peak signal to noise ratio (PSNR) [28] and structural similarity index metric (SSIM) [27] were used as quality evaluation indicators to evaluate performance. The larger the value of PSNR and SSIM, the better the image quality. All experiments were carried out on MATLAB R2020a running on Windows 10. The CPU was Intel Core i7-2600, the memory was 8 GB, and the frequency was 3.40 GHz.

### 5.1. Parameter Setting

The parameter settings involved in this algorithm are as follows. The size of the input matrix is m×n. In Algorithm 1, we set $A_{i}^{\left(0\right)}=0$, $B_{i}^{\left(0\right)}=0$. In Algorithm 2, we set $X_{i}^{\left(0\right)}=0$, $S_{i}^{\left(0\right)}=0$, $H_{i}^{\left(0\right)}=0$. Following the practice in [29] and our tests, we set the Lagrange multiplier matrices $Y_{1}^{\left(0\right)}=M/\max\left(∥M∥,\lambda^{-1}∥M{∥}_{\infty }\right)$, and $Y_{2}^{\left(0\right)}=0$, where ∥M∥ is the spectral norm of matrix *M* and ${∥M∥}_{\infty }$ is the maximum absolute value of the entries in matrix *M*. In addition, the constants $\epsilon_{X_{i}}$ and $\epsilon_{S_{i}}$ in Algorithm 2 are set to 0.1, the same as in [21].

In addition, we set θ in (Equation 13), which controls the sensitivity of the model to coefficient errors to be $1.25/\sqrt{\max(m,n)}$, and we set β in (Equation 11), which controls the sensitivity of the model to the TV norm to be $10^{-8}/\sqrt{\max(m,n)}$. Finally, in Algorithm 2, γ can be computed by γ=η/β, and μ corresponds to parameter $\mu^{\left(t\right)}$ in Algorithm 2, and we set δ=0.1.

The above parameters can be set according to experience. However, in Algorithm 1, the size of patch *b*, the number of non-local similar patches *m*, balancing factors ρ and η in (Equation 15), regularization parameters λ in (Equation 15) and τ in (Equation 17), and other parameters are determined by parameter experiments.

In the parameter experiments of patch size *b* and patch number *m*, the value range of *b* is multiples of 2 from 8 to 64, the value range of *m* is 100∼180, and the sampling interval is 20. Figure 5 shows the influence of the values of *b* and *m* on the restoration results when the noise density p=0.3. Therefore, according to the experiment, when the noise density is *p* = 0.3, we set the size of image blocks to 64×64 and look for 160 similar image patches. In addition, through experiments, when *p* = 0.2, *p* = 0.4, we set patch sizes to 60×60 and 70×70, and the number of non-local similar patches selected is 160 and 190.

It can be seen that when the noise density increases, the size of the image patch also increases. When the noise density increases, the original similar patches may be different from each other, so expanding the size of similar patches and increasing the number of similar patches can improve the accuracy of the algorithm.

In addition, we analyze the parameter experiments of balancing factors ρ and η, and regularization parameters λ and τ. Figure 6 shows that when the noise density p=0.3, the influence of the value of this parameter on the restoration result when other parameters are fixed is noticeable. We set ρ=1, η=0.1, λ=0.02, τ=0.5 when the noise density p=0.3.

In Algorithm 1, the maximum number of iterations maxiter, and in Algorithm 2, the internal maximum number of iterations inneriter, are parameters related to the convergence of the algorithm. Because SDWLR-GSC is a non-convex model, it is difficult to prove the global convergence of the algorithm in theory. Therefore, this paper analyzes the influence of iteration times on the restoration results through experiments. Through experiments, maxiter=2 and inneriter=100 are selected to avoid unnecessary iterative calculation.

### 5.2. Large Sparse Additive Noise Removal

We first test the experimental results of the proposed SDWLR-GSC for removing large and sparse additive noise. We suppose that the ratio of damaged pixels to all pixels in the image is *p*, and the values of these noisy pixels are 255, and a noisy observation image is generated. In this test, we give the denoising results of three noise levels, namely p=0.2, p=0.3, and p=0.4. There are several other parameters in the algorithm, which are set as follows. In Algorithm 1, the number of iterations *k* and the patch size are set according to the noise level. For higher noise levels, we choose larger patches and run more iterations. According to experiments, let the ratio of the number of damaged pixels to the number of all pixels be *p*, and set patch sizes to 60×60, 64×64, and 70×70 for p=0.2, p=0.3, and p=0.4, respectively. At these noise levels, the number of non-local similar patches selected is 160, 160, and 190, respectively.

We compare the proposed SDWLR-GSC with the most advanced existing methods, including the PCP algorithm [30], reweighted $L_{1}$ algorithm [31], NSVT method [21], and SRLRMR algorithm [29]. We also compare it with the LR-GSC algorithm [15]. It can be seen from Table 1 that under all noise levels, compared with other competitive methods, the SDWLR-GSC proposed by us achieves higher peak signal-to-noise ratio results. It can be seen from Table 1 that under all noise levels, compared with other competitive methods, the SDWLR-GSC proposed by us achieves higher PSNR results. In addition, the LR-GSC is more suitable for removing Gaussian noise than large and sparse noise.

When p=0.4 noise probability, on average, the PSNR performance of this method is 13.11 dB higher than that of PCP, 7.87 dB higher than that of specific gravity reweighted $L_{1}$, 11.97 dB higher than that of NSVT, and 5.09 dB higher than that of SRLRMR. Figure 7 shows the visual comparison results of an image (Lena) when the noise density is p=0.2. It can be observed that the PCP, reweighted $L_{1}$, and NSVT methods cannot completely restore the damaged image, while the SRLRMR method can better complete the image denoising task, but it is slightly lacking in structure and the edge is still blurred. In contrast, our proposed SDWLR-GSC method not only effectively eliminates noise, but also retains sharp edges and fine details.

### 5.3. Salt and Pepper Noise Removal

In this subsection, we apply the proposed SDWLR-GSC to remove salt and pepper noise—that is, we add the noise with a pixel value of 0 on the basis of the noise in the previous section, and then the ratio of damaged pixels’ total number with pixel values of 0 and 255 to all pixels’ number is *p*. Similarly, in this test, we still give three different noise levels of p=0.2, p=0.3, and p=0.4. We compare this model with LR-GSC [15], PCP [30], the WNNM-RPCA model [20], the WSNM-RPCA model [32], WSNM-$L_{1}$ [33], and the DWLP model [34].

Table 2 quantitatively compares the denoising results of various methods under different salt and pepper noise probability *p*. It can be seen that when *p* = 0.2, *p* = 0.3, *p* = 0.4, the PSNR of our model is higher than that of LR-GSC, PCP, WNNM-RPCA, WSNM-$L_{1}$, and DWLP. Moreover, Figure 8 shows the visual comparison results of an image (House) when the noise density is p=0.3.

Through the experiments, we can draw several conclusions: firstly, the strong salt and pepper noise destroys the sparse prior and low-rank of the image, and the restoration performance of the PCP model is poor; secondly, the PSNR of our model is higher than the average level of other models, which shows that using the dual-weighted model in group sparse coding, processing low-rank components and sparse components at the same time, and introducing the TV norm to solve the problem of image structure smoothing can reconstruct the low-rank structure of images more accurately.

## 6. Conclusions

In this paper, we constructed a smooth dual-weighted low-rank group sparse coding model. It combined group sparse coding, the TV norm, and a dual-weighted model. We also proved the superior performance of the proposed method in image denoising. Experimental results show that this method is obviously superior to the original PCP optimization, reweighted $L_{1}$ norm minimization, and NSVT and SRLRMR algorithms in removing large and sparse additive noise, and it is obviously superior to the PCP optimization, WNNM-RPCA, WSNM-RPCA, WSNM-$L_{1}$, and DWLP methods in removing salt and pepper noise.

Although our proposed method has good performance, there is still room for further improvement. In this paper, the algorithm solver is based on the alternating direction method, due to the matrix–matrix multiplications and matrix inversions, and singular value decomposition is required for each iteration, which has high computational complexity for large matrices. In addition, when the expectation matrix becomes complex—for example, it has a high internal rank structure or the deletion becomes dense—satisfactory performance may not be obtained. Therefore, the question of how to reduce the computational complexity while maintaining high performance will be our research direction in the future. In addition, other applications of the proposed method are another important issue for future work.

## Figures and Tables

**Figure 1 entropy-24-00946-f001:**
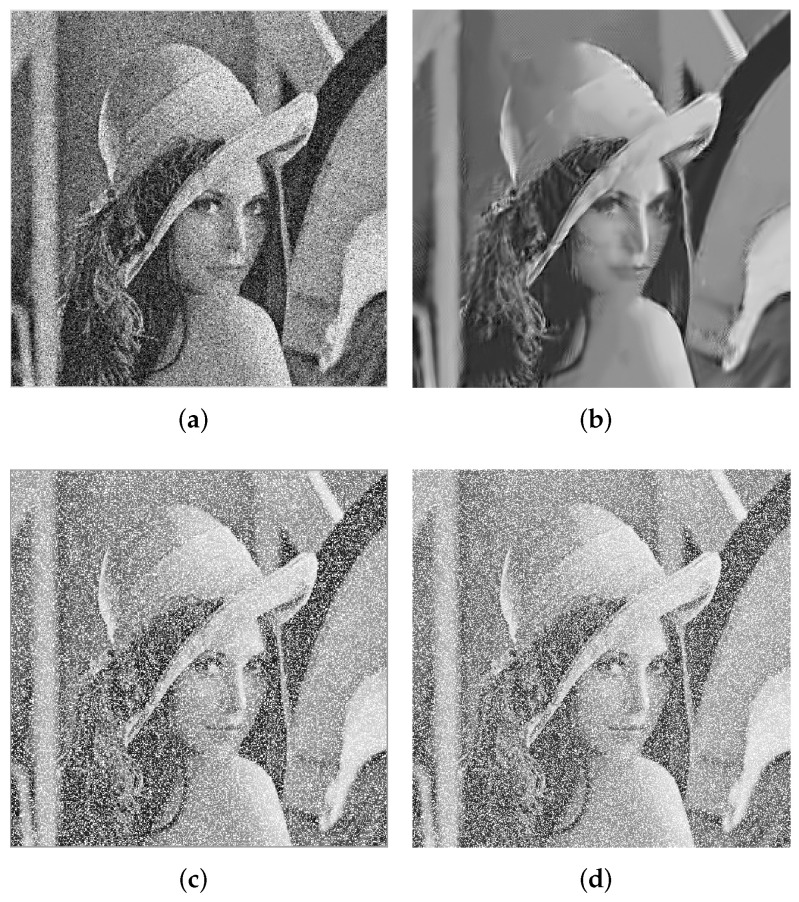
(**a**) The Lena image corrupted by Gaussian noise. (**b**) Restoration result of (**a**) based on LR-GSC. (**c**) The Lena image corrupted by outlier noise. (**d**) Restoration result of (**c**) based on LR-GSC.

**Figure 2 entropy-24-00946-f002:**
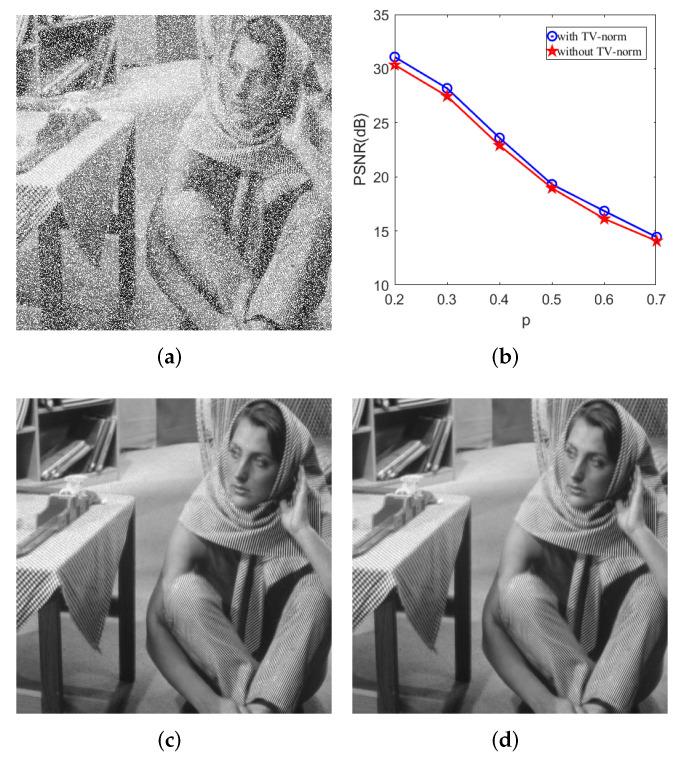
Comparison of the denoising results with TV norm and without; 30% of pixels of Barbara image are corrupted by large and sparse noise. (**a**) The corrupted image. (**b**) PSNR values for Barbara image corrupted by large and sparse noise with different density. (**c**) Restored by the model of dual-weighted low-rank group sparse coding without TV norm. PSNR = 30.34 dB. (**d**) Restored by the model of dual-weighted low-rank group sparse coding with TV norm. PSNR = 31.08 dB.

**Figure 3 entropy-24-00946-f003:**
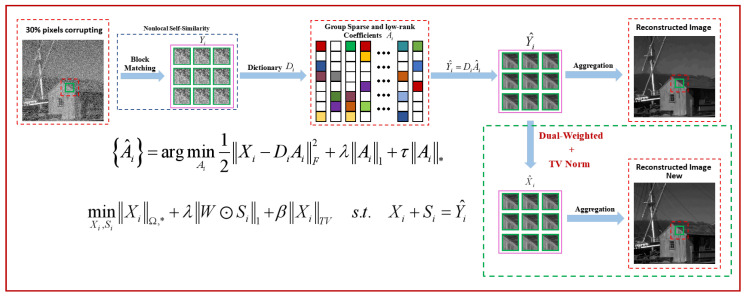
Flowchart of the proposed SDWLR-GSC model for image denoising. On the basis of LR-GSC [15], we applied the dual-weighted model to the reconstructed similar patches, and introduced the TV norm to maintain the smoothness of the image structure. Finally, the optimal sparse codes are used to estimate the clean patch groups for constructing the restored image. (Firstly, the corrupted image is extracted from non-local similar patches through a block matching operator. Secondly, patches with similar structures are grouped to perform dictionary learning to obtain group coefficients. At the same time, the group coefficients remain sparse and low-rank concurrently).

**Figure 4 entropy-24-00946-f004:**
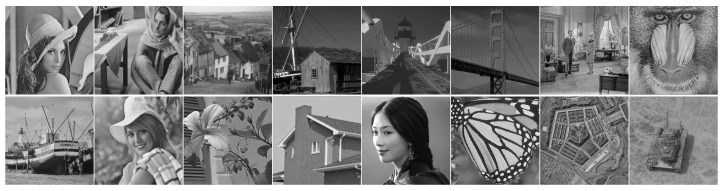
Test images in our experiments. First row: Lena, Barbara, Goldhill, Frame, Road, Bridge, Couple, Monkey. Second row: Boat, Elaine, Flower, House, Lin, Monarch, Pentagon, Tank.

**Figure 5 entropy-24-00946-f005:**
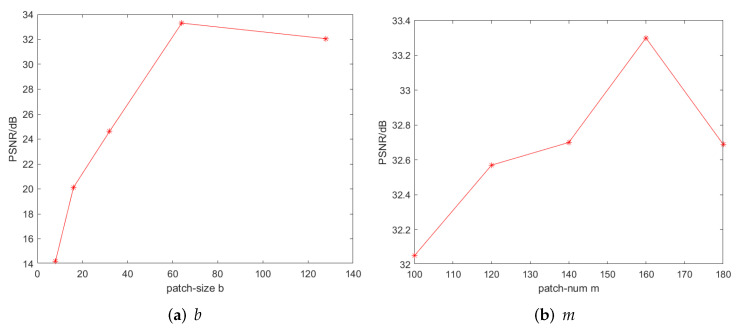
The influence of the values of *b* and *m* on the restoration results when the noise density p=0.3.

**Figure 6 entropy-24-00946-f006:**
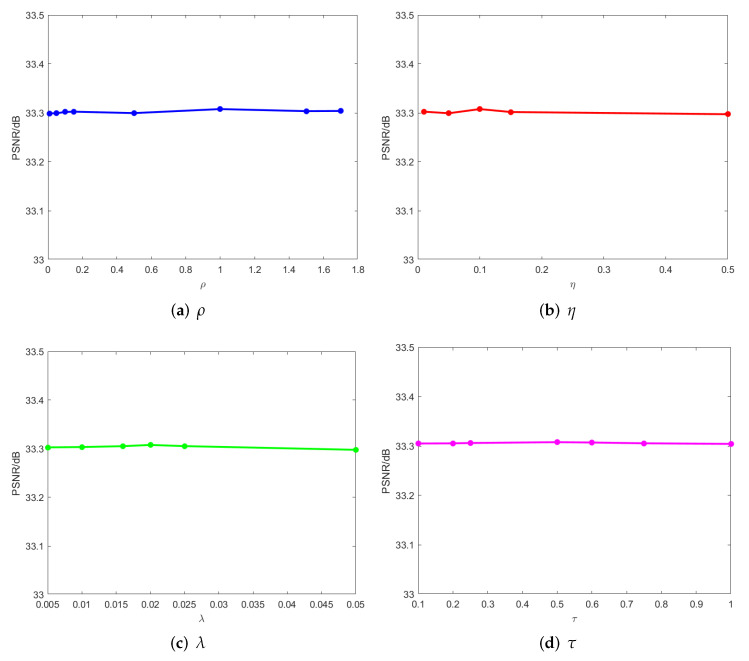
The influence of the values of ρ, η, λ, and τ on the restoration results when the noise density p=0.3.

**Figure 7 entropy-24-00946-f007:**
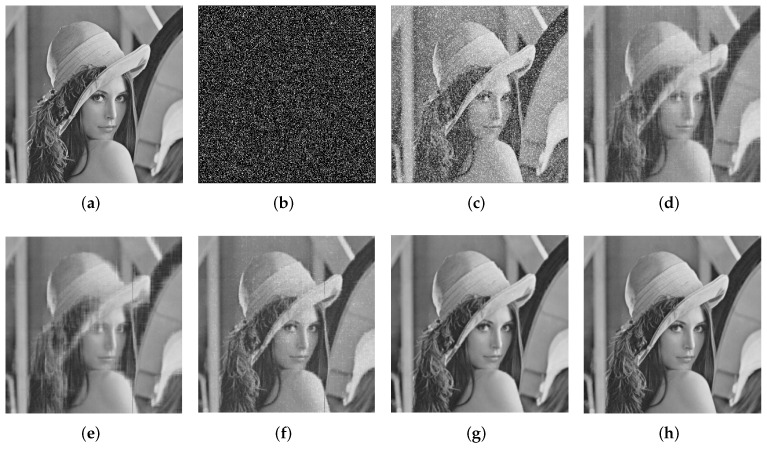
Restoration results comparison of the single image, Lena, of the PCP, reweighted l1, NSVT, SRLRMR, and our method. Here, 20% image pixels are corrupted by large and sparse noise. (**a**) The original Lena image. (**b**) The large and sparse noise. (**c**) The corrupted image (12.24 dB). (**d**) Restoration result by PCP (26.75 dB). (**e**) Restoration result by reweighted $l_{1}$ norm minimization (23.84 dB). (**f**) Restoration result by NSVT (32.15 dB). (**g**) Restoration result by SRLRMR (32.47 dB). (**h**) Restoration result by our method (36.76 dB).

**Figure 8 entropy-24-00946-f008:**
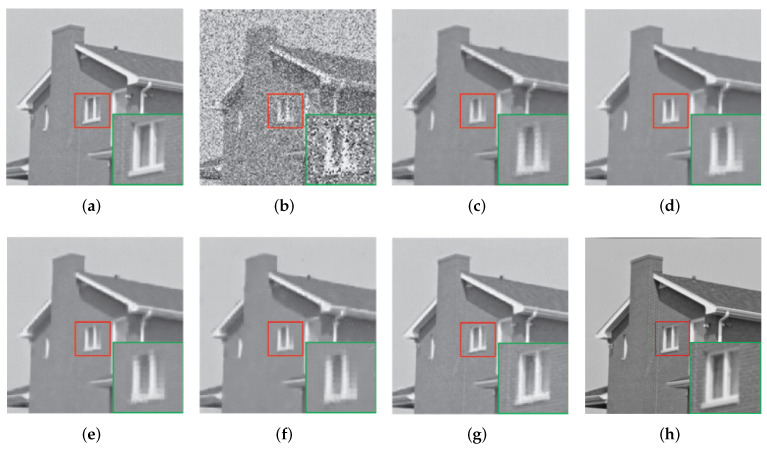
Restoration results comparison of the image House of the PCP, WNNM-RPCA, WSNM-RPCA, WSNM-$l_{1}$, DWLP, and our method. Here, 30% image pixels are corrupted by salt and pepper noise. (**a**) The original Lena image, and the image size is 512 × 512. (**b**) The input corrupted image (10.69 dB). (**c**) Restoration result by PCP (28.88dB). (**d**) Restoration result by WNNM-RPCA (29.51 dB). (**e**) Restoration result by WSNM-RPCA (29.77 dB). (**f**) Restoration result by WSNM-RPCA $l_{1}$ norm minimization (28.15dB). (**g**) Restoration result by DWLP (33.89 dB). (**h**) Restoration result by our method (38.08 dB).

**Table 1 entropy-24-00946-t001:** Comparison of different denoising methods under different large sparse noise probabilities in terms of PSNR.

*p* Value	Image No.	PSNR (dB)					
		LR-GSC	PCP	Reweight $l_{1}$	NSVT	SRLRMR	SDWLR-GSC (Ours)
*p* = 0.2	Lena	12.25	26.75	23.84	32.15	32.47	**36.76**
	Barbara	11.74	24.23	22.33	26.28	26.36	**34.56**
	Goldhill	11.56	28.41	26.75	31.99	33.71	**34.48**
	Frame	9.63	25.90	22.79	28.10	29.64	**30.95**
	Road	10.01	29.77	26.43	34.63	**34.88**	34.50
	Bridge	10.17	29.98	27.96	34.01	33.73	**34.69**
	Couple	12.19	25.71	23.49	28.78	29.31	**31.18**
	Monkey	12.67	21.72	20.43	22.12	23.71	**25.48**
	Boat	12.86	25.22	23.01	27.65	28.35	**32.23**
	**Average**	11.45	26.41	24.11	29.52	30.24	**32.76**
*p* = 0.3	Lena	10.48	21.27	22.17	23.84	30.67	**33.31**
	Barbara	9.97	19.10	20.85	20.77	24.35	**31.08**
	Goldhill	9.80	23.00	25.42	25.56	31.15	**31.91**
	Frame	7.86	22.52	23.04	26.61	27.36	**29.31**
	Road	8.24	25.20	26.69	28.73	**33.31**	33.18
	Bridge	8.41	27.29	26.57	30.57	33.01	**33.09**
	Couple	10.44	20.92	22.01	22.93	27.60	**28.18**
	Monkey	10.91	18.09	19.21	18.12	21.90	**22.93**
	Boat	11.12	21.25	21.62	22.01	27.66	**28.74**
	**Average**	9.69	22.07	23.06	24.35	28.56	**30.19**
*p* = 0.4	Lena	9.22	13.18	16.53	13.57	20.53	**27.44**
	Barbara	8.71	12.25	15.10	12.50	17.88	**24.43**
	Goldhill	8.54	13.04	19.39	13.80	21.66	**27.48**
	Frame	7.61	12.32	19.61	15.69	22.58	**26.26**
	Road	6.98	11.39	18.65	13.21	20.04	**29.87**
	Bridge	7.14	13.94	24.49	18.04	28.12	**32.15**
	Couple	9.18	13.28	17.24	13.50	20.55	**23.60**
	Monkey	9.65	12.65	15.00	12.06	17.07	**18.68**
	Boat	9.86	13.82	17.03	13.84	19.64	**24.01**
	**Average**	8.54	12.88	18.12	14.02	20.90	**25.99**

**Table 2 entropy-24-00946-t002:** Comparison of different denoising methods under different salt and pepper probabilities in terms of PSNR.

*p* Value	Image	PSNR (dB)						
		LR-GSC	PCP	WNNM-RPCA	WSNM-RPCA	WSNM-L1	DWLP	SDWLR-GS (Ours)
*p* = 0.2	Couple	12.60	26.34	26.03	26.29	25.08	31.57	**31.73**
	Elaine	12.58	30.35	30.24	30.48	28.64	36.38	**42.21**
	Flower	12.64	27.94	27.46	27.67	26.59	31.78	**41.30**
	Goldhill	12.46	28.64	28.00	28.16	26.79	32.97	**34.09**
	House	12.56	30.29	31.37	31.43	29.84	37.19	**46.21**
	Lin	12.01	29.61	29.51	29.54	26.72	33.04	**38.52**
	Monarch	12.15	24.51	26.38	26.55	25.16	29.53	**41.30**
	Pentagon	12.72	25.84	25.51	25.66	24.29	31.20	**31.83**
	Tank	12.91	31.67	30.12	30.23	29.49	**35.92**	34.57
	**Average**	12.51	28.35	28.29	28.45	26.96	32.29	**37.97**
*p* = 0.3	Couple	10.88	24.50	24.41	24.64	23.91	28.77	**29.68**
	Elaine	10.80	28.42	28.44	28.65	27.15	33.46	**38.77**
	Flower	10.91	27.11	26.28	26.43	25.38	29.29	**36.25**
	Goldhill	10.71	27.43	26.56	28.16	25.63	30.30	**31.65**
	House	10.76	28.88	29.51	29.77	28.15	33.89	**38.08**
	Lin	10.29	27.22	27.80	27.93	26.18	30.58	**36.03**
	Monarch	10.40	22.84	24.27	24.47	23.61	26.39	**36.82**
	Pentagon	10.95	24.15	23.97	24.18	23.16	28.93	**29.45**
	Tank	11.14	30.05	29.57	29.64	28.57	**33.21**	32.73
	**Average**	10.76	26.73	26.76	27.10	25.75	30.54	**34.97**
*p* = 0.4	Couple	9.60	23.43	23.77	23.81	22.62	26.57	**28.35**
	Elaine	9.55	25.54	26.60	27.03	25.18	30.71	**35.94**
	Flower	9.64	25.78	25.68	25.60	24.23	28.03	**33.43**
	Goldhill	9.44	24.79	25.40	25.92	24.12	28.68	**30.22**
	House	9.52	24.83	27.90	27.92	25.97	31.04	**34.60**
	Lin	9.02	25.43	26.11	26.15	23.71	27.23	**33.97**
	Monarch	9.14	21.34	22.98	23.22	22.22	24.32	**34.42**
	Pentagon	9.73	22.34	23.04	22.96	21.95	26.13	**27.97**
	Tank	9.90	27.57	29.12	29.19	27.29	**31.94**	31.70
	**Average**	9.50	24.56	25.62	25.76	24.14	28.29	**32.29**

## Data Availability

Not applicable.

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
