# Peer review of "Structural Smoothing Low-Rank Matrix Restoration Based on Sparse Coding and Dual-Weighted Model"

_entropy, 2022, doi:10.3390/e24070946_

Round 1

Reviewer 1 Report

The authors proposed a method for image restoration based on low-rank regularized group sparse coding and dual weighting. The manuscript is a incremental work of the following paper:

Zha, Zhiyuan, et al. "Image restoration via reconciliation of group sparsity and low-rank models." IEEE Transactions on Image Processing 30 (2021): 5223-5238. The paper may be published upon fully addressing following issues:

1) There is no citation in the first paragraph of introduction section. Each new facts and observations should be cited.

2) line 49: “As the two-dimensional extension of sparse representation theory, low-rank matrix theory also has its sparse representation model, which is called low-rank representation.” It is not technically meaningful

3) line 78: 1)We introduce a group sparse coding model into the low-rank model. Using the non-local self similarity prior of the image, the proposed method can effectively find and remove noise, and enhance the structure and regional smoothness.

it is not a contribution. it is proposed in :

Zha, Zhiyuan, et al. "Image restoration via reconciliation of group sparsity and low-rank models." IEEE Transactions on Image Processing 30 (2021): 5223-5238.

4) line 93: #3 is not a contribution.

5) line 127: sum of matrix elements ==> sum of matrix absolute elements

6) figure 3 and its caption are taken from :

Zha, Zhiyuan, et al. "Image restoration via reconciliation of group sparsity and low-rank models." IEEE Transactions on Image Processing 30 (2021): 5223-5238.

So, the credit should be given to original work by proper citation.

7 ) Figure 4 only has two rows, however the caption describes three rows

8) line 278: the actual reference for SRLRMR algorithm is the following work:

Wang, Hengyou, et al. "Reweighted low-rank matrix analysis with structural smoothness for image denoising." IEEE Transactions on Image Processing 27.4 (2017): 1777-1792.

which is wrongly cited in the manuscript.

9) The last column of table 1 is not informative. it can be removed.

10 ) The following work should be added to baseline algorithms:

Zha, Zhiyuan, et al. "Image restoration via reconciliation of group sparsity and low-rank models." IEEE Transactions on Image Processing 30 (2021): 5223-5238.

11) The English should be improved. There are too many errors. For example:

line 18: Among these work ==> Among these works

line 121: [18] has been proved ==> authors in [18] proved

Author Response

We thank Reviewer 1 for the careful review and insightful comments!

Comment 1: There is no citation in the first paragraph of introduction section. Each new facts and observations should be cited. 

Response: Thanks for this valuable suggestion! In the revised manuscript, we have cited Reference [1] to [4] in the first paragraph of introduction section as the reviewer suggested.

Comment 2: line 49: As the two-dimensional extension of sparse representation theory, low-rank matrix theory also has its sparse representation model, which is called low-rank representation. It is not technically meaningful. 

Response: Thanks for this suggestion! In the revised manuscript, we have deleted this sentence.

Comment 3: line 78: 1)We introduce a group sparse coding model into the low-rank model. Using the non-local self similarity prior of the image, the proposed method can effectively find and remove noise, and enhance the structure and regional smoothness.it is not a contribution. it is proposed in :Zha, Zhiyuan, et al. "Image restoration via reconciliation of group sparsity and low-rank models." IEEE Transactions on Image Processing 30 (2021): 5223-5238. 

Response: Sorry for this confusion. We have rewritten the line 78. What we want to illustrate is that on the basis of maintaining the low-rank of similar patch coefficients when LR-GSC constructs similar patches, we also maintain the low-rank of similar patch reconstruction matrix.

In this manuscript, we not only consider the low-rank property between similar patches, but also combine the model with TV norm to further strengthen the local structural smoothness of the matrix, which is our main contribution of this work.

Comment 4: line 93: #3 is not a contribution. 

Response: Thanks for this suggestion! In the revised manuscript, we have deleted this contribution and maintain the others.

Comment 5: line 127: sum of matrix elements ==> sum of matrix absolute elements. 

Response: Thanks for this valuable suggestion! In the revised manuscript, we have changed "sum of matrix elements" to "sum of matrix absolute elements".

Comment 6: figure 3 and its caption are taken from :Zha, Zhiyuan, et al. "Image restoration via reconciliation of group sparsity and low-rank models." IEEE Transactions on Image Processing 30 (2021): 5223-5238. So, the credit should be given to original work by proper citation. 

Response: Thanks for this suggestion! In the revised manuscript, we have cited this work. In addition, the flowchart in Figure 3 is redrawn by us on the basis of LR-GSC, which adds the ideas of our algorithm.

Comment 7: Figure 4 only has two rows, however the caption describes three rows. 

Response: We are sorry for such a mistake. In the revised manuscript, we have modified it correctly.

Comment 8: line 278: the actual reference for SRLRMR algorithm is the following work:Wang, Hengyou, et al. "Reweighted low-rank matrix analysis with structural smoothness for image denoising." IEEE Transactions on Image Processing 27.4 (2017): 1777-1792. which is wrongly cited in the manuscript. 

Response: Thanks for your detailed review! We are so sorry for our confusion. In the revised manuscript, we have revised it to the correct reference.

Comment 9: The last column of table 1 is not informative. it can be removed. 

Response: Thanks for your suggestion! In the revised manuscript, we have deleted the last column of table 1.

Comment 10: The following work should be added to baseline algorithms:

Zha, Zhiyuan, et al. "Image restoration via reconciliation of group sparsity and low-rank models." IEEE Transactions on Image Processing 30 (2021): 5223-5238. 

Response: Thanks for your valuable suggestion! In the revised manuscript, we have added the LR-GSC to baseline algorithms. As said in the work of this reference, the LR-GSC method has a better performance for image restoration with Gaussian noise, but it is vulnerable for outlier noise. Thus, inspired by LR-GSC, we mainly propose SDWLR-GSC method and address the problem of image restoration with outlier noise. And finally, our proposed method has improved the performance significantly.

Comment 11: The English should be improved. There are too many errors. For example:line 18: Among these work ==> Among these works line 121: [18] has been proved ==> authors in [18] proved 

Response: Thank you for your valuable and thoughtful comments. We are sorry for such mistakes. In the revised manuscript, we have changed "these work" to "these works" and “[18] has been proved” to “authors in [18] proved”. And we have carefully checked and improved the English writing in the revised manuscript. Thanks again.

Reviewer 2 Report

The manuscript addresses image restoration in the presence of outlier noise, formulating a novel smoothing dual-weighted low-rank group sparse coding algorithm. The methodology is made more efficient through the use of inexact augmented Lagrange multipliers and non-uniform singular value thresholds, demonstrating improved results with regard to peak signal to noise ratio in comparison to other modern image restoration algorithms.

The paper is well organized, appropriately reviews the existing literature, and appears scientifically sound. Figures and tables are clearly presented, well selected, and sufficiently detailed. I have only a few minor suggestions for improvement:

-The conclusions section helpfully discusses some limitations of the proposed methodology. This is indeed important to include, but presents as too vague compared to many of the more detailed parts of the paper.

- There are recurrent grammatical issues throughout the paper (subject-verb agreement, missing articles); this should be carefully addressed to ensure readability.

-In the abstract, the final sentence is unclear and should be adjusted.

Author Response

We thank Reviewer 2 for the careful review and insightful comments! Comment 1: The conclusions section helpfully discusses some limitations of the proposed methodology. This is indeed important to include, but presents as too vague compared to many of the more detailed parts of the paper. 

Response: Thanks for your valuable suggestion! In the revised manuscript, we have rewritten the limitations of the proposed methodology in the conclusions as the reviewer suggested.

Comment 2: There are recurrent grammatical issues throughout the paper (subject-verb agreement, missing articles); this should be carefully addressed to ensure readability. 

Response: Thank you for your valuable and thoughtful comments. We are sorry for such mistakes. We have carefully checked and improved the English writing in the revised manuscript.

Comment 3: In the abstract, the final sentence is unclear and should be adjusted. 

Response: Thanks for your detailed review! Sorry for the confusion. We have rewritten the final sentence in the abstract.

Reviewer 3 Report

Nowadays, significant advances in image restoration/reconstruction have been made, dominated by neural network, CNN, and transformers ..
In all your paper no one description, comparison, discussion with the deep models has been done.
Although your article is well written and fully detailed, a modification of the error function with the addition of a weight (dual weighted in your case) is not enough. A CNN can have thousand or million of parameters and the researcher analyze the behaviour of the new layers introduced and the relative connections (e.g skip connections, specific convolutive layers, new loss functions).
I think your direction is wrong to try to find local descriptor setting many parameters and considering using PCA (a technique around 1920's). However, I strongly suggest you apply an image-restoration model and report the results using the same metric (psnr) as reported in your paper and compare your dual weight proposal with some deep models, such that to enforce your description. You can report the computational time for both of them and, improve the literature by introducing different deep models.

Author Response

Thank you very much for your valuable and thoughtful comments. In this manuscript, we mainly address the image restoration/reconstruction problem by traditional optimal model based on low-rank theory, which is also an interesting and meaningful research issue. Nowadays, although significant advances in image restoration/reconstruction have been made by neural network, CNN, and transformers, but it is hard to be understood. Thus, traditional image restoration methods should also be concerned by researchers.

Round 2

Reviewer 1 Report

The authors addressed all my concerns. I have no more comments.

Reviewer 2 Report

The authors have well addressed my concerns at this point.

Reviewer 3 Report

ok